# Positive Childhood Experiences Associate with Adult Flourishing Amidst Adversity: A Cross Sectional Survey Study with a National Sample of Young Adults

**DOI:** 10.3390/ijerph192214956

**Published:** 2022-11-13

**Authors:** Zhiyuan Yu, Lin Wang, Wenyi Chen, Juan Zhang, Amie F. Bettencourt

**Affiliations:** 1School of Nursing, Johns Hopkins University, Baltimore, MD 21205, USA; 2School of Nursing, Shanghai Jiao Tong University, Shanghai 200025, China; 3International Peace Maternity and Children Hospital of China Welfare Institution, Departments of Nursing, School of Medicine, Shanghai Jiao Tong University, Shanghai 200025, China; 4Department of Psychiatry and Behavioral Sciences, School of Medicine, Johns Hopkins University, Baltimore, MD 21205, USA

**Keywords:** adverse childhood experiences, flourishing, positive childhood experiences, surveys, well-being, young adults

## Abstract

The purpose of this study was to examine the prevalence of PCEs among young adults in Mainland China and the extent to which the cumulative number of PCEs moderates the associations between ACEs and flourishing in adulthood. Between August and November 2020, we used convenience and snowball sampling to recruit 9468 young adults, ages 18–35, enrolled in undergraduate or graduate programs at universities in Mainland China to participate in a survey, which included measures on flourishing, exposure to ACEs and PCEs, and demographic characteristics. Approximately 92% of participants reported experiencing seven to nine PCEs, with harmonious family relationships (96.9%), feeling supported by friends (96.8%) and being treated fairly at school (96.3%) being the most common PCEs reported. Results of the multiple regression indicated that the cumulative number of PCEs statistically significantly moderated the relation between the cumulative number of ACEs and flourishing (interaction term *b* = −0.060 [−0.071, −0.049], *p* < 0.001, adjusted R^2^ = 0.183); as the number of ACEs increased up through eight ACEs, decreases in flourishing were smaller among those with higher numbers of PCEs. PCEs are common among young adults from Mainland China and serve a potential buffering effect against exposure to ACEs.

## 1. Introduction

Adverse childhood experiences (ACEs) are potentially traumatic experiences in the first 18 years of life, such as abuse, neglect, and living in a stressful household or community environment (e.g., exposure to domestic or community violence, living with someone with mental illness, or bullying) [1]. In the United States, approximately 62% of adults and 48% of children have at least one ACE [2,3]. Since the landmark CDC-Kaiser Permanente ACE study in 1998, it has been well-established that ACEs have deleterious, long-lasting impact on individuals’ health and well-being [1,4,5,6,7]. Nevertheless, many people exposed to ACEs do not develop poor health outcomes [8,9,10,11]. Empirical evidence suggests that positive childhood experiences (PCEs) can co-occur with ACEs [12] and may buffer the impact of ACEs on health and well-being [13].

### 1.1. Positive Childhood Experiences

PCEs are “experiences before age 18 that are thought to be beneficial, such as positive relationships with parents and other adults, household routines, beliefs that provide comfort, and having good neighbors” [9,14]. In the literature, terms such as positive childhood experiences (PCE), benevolent childhood experiences (BCEs), advantageous childhood experiences, counter-ACEs, or protective and compensatory experiences (PACEs) are often used interchangeably to examine the cumulative impact of positive experiences in childhood [8,11,13,15,16]. Unlike ACEs, the prevalence of PCEs has less frequently been studied. In a statewide sample of over 6000 adults in Wisconsin, U.S., it was estimated that 13.2% reported zero to two PCEs, 34.5% reported three to five PCEs, and 52.3% reported six to seven PCEs [13]. Baglivio and Wolff (2021) measured PCEs in a sample of 28,048 juvenile offenders in Florida, U.S., using the 11 specific protective exposures extrapolated from the Positive Achievement Change Tool (PACT) assessment, and found that 68.0% of participants reported less than six PCEs and 32.0% of participants reported six or more PCEs [17]. These findings suggest that positive childhood experiences are common, even amid adversity.

### 1.2. The Protective Role of PCEs

An emerging body of literature has demonstrated the long-term benefits of PCEs on health and well-being, such as reducing the risks of developing poor mental health (e.g., depression and anxiety) in adolescence and adulthood [13,18,19,20]; promoting adult relational health [13]; positive functioning [21]; and cardiovascular health in midlife [22]; and lowering the level of risky reproductive planning and prenatal stressful life events [23]. PCEs are also modifiable protective factors and can be an important source of resilience that buffers the deleterious effects of ACEs on health outcomes, such as childhood obesity [24], adolescent pregnancy [25], risky behaviors [26], mental health conditions [13,27,28], and educational attendance [29,30].

In a new conceptualization of ACEs and resilience, Weems et al. (2021) proposed an integrative term, traumatic and adverse childhood experiences (TRACES+), which acknowledges both risk and protective factors that contribute to equifinality (i.e., multiple different risks may lead to the same negative outcomes) and multifocality (i.e., a particular risk may have heterogenous outcomes) [31]. This integrative conceptualization adds a resilience pinnacle to the traditional ACEs pyramid and emphasizes that there are multiple levels of interventions possible to direct individuals on a path of resilience [31]. This TRACE+ serves as a conceptual model for this study where we hypothesize that PCEs are some of the protective factors that mitigate the effects of exposure to ACEs and can place individuals on the path towards the pinnacle of resilience.

### 1.3. Knowledge Gap and Relevance to Public Health

Even though PCEs co-occur with and may buffer the adversarial effects of ACEs, research to date has primarily focused on the impact of ACEs on poor health and developmental outcomes without considering PCEs. This approach provides a skewed view of childhood experiences and underappreciates the protective role of positive experiences (e.g., sensitive, responsive parenting) in health and developmental outcomes [12,32]. Simultaneously assessing ACEs and PCEs can support identifying aspects of developmental pathways that are significant for etiologic outcomes and amenable to intervention [33]. Understanding the impact of ACEs along with PCEs may also help explain heterogeneity in outcomes in populations, particularly those exposed to substantial adversity.

Resilience theory is a conceptual framework that provides a strengths-based approach to understanding child and adolescent development and informing intervention design [34]. Resilience theory focuses on promotive factors that are positive contextual, social, and individual variables that buffer the negative effects of risk exposure. Consistent with resilience theory, existing research on PCEs in the context of ACEs often focuses on positive contextual, social, and individual variables that interfere or disrupt developmental trajectories from risk to problem behaviors, mental distress, and poor health outcomes [34]. The knowledge of the mechanisms through which PCEs interrupt ACEs’ physiological/behavioral/psychosocial effects remains limited but there are some hypothesized pathways. For example, in a statewide sample of adults in Wisconsin (*n* = 9188), Bethell et al. (2019) found that PCEs (e.g., felt safe and protected by an adult in their home) had a dose-response association with adult depression and/or poor mental health and adult-reported social and emotional support (ARSES) after adjustment for ACEs [13]. They also found that PCEs associations with adult mental health outcomes remained significant and changed modestly when ARSES were included. Thus, one hypothesized behavioral mechanism through which PCEs interrupt ACEs effects is that PCEs may promote positive health such as acquiring social and emotional support in adulthood, which, in turn, may reduce health burden associated with ACEs [13].

Although informative in providing evidence on protective factors for disease prevention in the face of adversity, precisely how PCEs and ACEs jointly shape broader health outcomes and well-being remains unclear. Understanding how PCEs shape health outcomes in the face of adversity would provide evidence for health professionals and policy makers to make informed decisions concerning the development and allocation of health promotion resources and preventative intervention services that not only buffer adversity but also promote resilience and well-being.

### 1.4. Human Flourishing as a Well-Being Indicator

Human flourishing is a comprehensive assessment of holistic well-being, and it is a complex and multidimensional concept that goes beyond psychological well-being [35,36]. Human flourishing comprises six domains: (1) happiness and life satisfaction; (2) physical and mental health; (3) meaning and purpose; (4) character and virtue; (5) close social relationships; and (6) financial and material stability [35]. These domains of human flourishing are often considered universally desirable and reflects the World Health Organization (WHO) definition of health, “a state of complete physical, mental and social well-being and not merely the absence of disease or infirmity” [37].

Research has found that childhood adversity is associated with lower levels of flourishing, while positive childhood experiences such as higher levels of family connection and parental warmth promote flourishing in children and adults [13,38,39,40,41,42,43]. Existing research on childhood experiences and flourishing has primarily been examined within Western contexts. How these relationships manifest in a non-Western context, such as mainland China (which is the most populous country in the world) is still not well understood.

### 1.5. ACEs, PCEs, and Flourishing in Chinese Young Adult

ACEs are common in mainland China. A cross-sectional survey study with 9468 Chinese young adults shows that approximately 56% reported at least one ACE, and 7% reported four or more ACEs and that higher exposure to ACEs is associated with lower levels of adult flourishing [44]. However, little is known about the prevalence and impact of PCEs in mainland China. In a cross-sectional sample of 6363 Chinese primary and secondary school students aged 8–18 years, 27.2% reported 0–2 PCEs, 46.4% reported 3–5 PCEs; and 26.4% reported 6–7 PCEs [19]. This study also found that PCEs may buffer the adverse impact of ACEs on adolescent depression and anxiety. Li and colleagues (2020) examined the relationship between childhood maltreatment and psychological flourishing in a sample of 1622 Chinese undergraduate students [45]. They found that childhood maltreatment was negatively associated with psychological flourishing. However, no study to date has explored the prevalence and impact of PCEs on flourishing in the context of ACEs among Chinese young adults.

### 1.6. Current Study

This cross-sectional, descriptive survey study aims to examine (1) the prevalence of PCEs among young adults in Mainland China and (2) the extent to which PCEs moderate the associations between ACE exposure and the level of flourishing in adulthood. We hypothesize that (a) greater exposure to PCEs will be associated with higher levels of adult flourishing; (b) greater exposure to PCEs will buffer the effect of ACEs on adult flourishing, such that ACEs will have a weaker association with adult flourishing for people exposed to higher levels of PCEs compared with those with lower PCEs exposure.

## 2. Materials and Methods

### 2.1. Study Procedure

Participants were recruited virtually through convenience and snowball sampling using an online survey platform from August to November 2020. The survey link was distributed via student cohorts’ online groups on WeChat, the most used communication software in Mainland China. The survey was anonymous and programmed to allow the same electronic device to complete the survey only once. At the end of the survey, participants were provided with online and community resources for mental health support and information on childhood adversities.

### 2.2. Sample

Young adults who were (a) 18 to 35 years old, as defined by the Erikson’s theory of psychosocial development [46], and (b) had enrolled in an undergraduate or graduate program at universities in Mainland China were eligible to participate. A total of 11,305 individuals responded to the survey. After excluding 676 ineligible individuals (e.g., less than 18 years old) and 1161 individuals with over 25% missing data on survey responses, we included a final sample of 9468 respondents in the data analysis. See Appendix A for comparisons on the observed characteristics between the final sample and those excluded due to missing data.

### 2.3. Measures

#### 2.3.1. Positive Childhood Experiences (PCEs)

PCEs were measured using the Chinese version of Positive Childhood Experiences Scale (C-PCEs). The C-PCEs included 9 items that asked respondents to self-report during the first 18 years of their lives how often or how much they: (1) felt able to talk to their family about feelings; (2) felt their family stood by them during difficult times; (3) felt safe and protected by an adult in their home; (4) felt their family relationships are harmonious; (5) felt treated fairly at school; (6) felt a sense of belonging in school; (7) felt supported by friends; (8) had at least two non-parent adults who took genuine interest in them; and (9) received affirmation, encouragement, or support. C-PCE items 1, 2, 3, 6, 7, and 8 were translated from the items in the Positive Childhood Experiences Scale developed by Bethell and colleagues [13]. C-PCE item 4, 5, and 9 were developed by the study team based on (1) findings from cognitive interviews with Chinese young adults with a purpose to elicit items that are culturally specific to the Chinese populations (e.g., item 4, harmonious family relationships [47,48]); and (2) factors that are associated with resilience in the child’s social ecology (e.g., item 5, treated fairly at school [49,50]).

The response options for these nine items of PCEs were “Never,” “Rarely,” “Sometimes,” “often,” and “Very often.” Consistent with the original Positive Childhood Experiences Scale [13], responses to each PCEs item were dichotomized into 0 and 1. In the original scale, responses were dichotomized into “0 = Never, rarely, or sometimes” and “1 = Very often or often”. However, we included “Sometime” as an affirmative response as well (i.e., “1 = Very often, often, or sometimes”) to account for the documented tendency among Chinese survey respondents to give more modest ratings of their own success and that of their parents, as modesty is a Confucius virtue valued in Chinese culture [51]. The total score of the scale range is 0–9, with higher scores indicating higher exposure to PCEs. The C-PCEs Scale demonstrated good content validity (Cronbach’s α = 0.72) and test-rest reliability (intraclass correlation [ICC] = 0.75) in this study sample. Factor analysis confirmed the C-PCEs comprised two subdimensions, household (items 1–4) and community (items 5–9) PCEs. The scores of PCEs subdimensions were the average of the items included within each dimension. Appendix A present detailed results of psychometric evaluations of the C-PCEs.

#### 2.3.2. Adverse Childhood Experiences (ACEs)

ACEs were measured using the adapted Chinese version of the World Health Organization (WHO) ACE-International Questionnaire (C-ACE-IQ [52]). The C-ACE-IQ assesses 12 categories of childhood adversities: (1) physical abuse, (2) emotional abuse, (3) sexual abuse, (4) family substance abuse, (5) incarcerated household member, (6) family mental illness, (7) household violence, (8) parental separation or divorce, (9) emotional neglect, (10) physical neglect, (11) bullying, and (12) community violence. Each category may have multiple items and responses to each item may include binary answers (i.e., “Yes” or “No”) or frequency answers (e.g., “Many times,” “A few times,” “Once,” Or “Never”). Following the scoring recommendation of the original WHO ACE-IQ questionnaire [53], each category of ACEs was dichotomized into non-exposure (scored 0) and exposure (scored 1). Thus, the cumulative ACEs score ranges from 0–12, with higher scores indicating higher reported exposure to ACEs. The C-ACE-IQ demonstrated good content validity (scale content validity index [S-CVI] = 0.89), and test-retest reliability (ICC = 0.88) in a sample of Chinese university students (*n* = 566) [52].

#### 2.3.3. Flourishing

Flourishing in adulthood was assessed using the Chinese version of the Flourishing Measure [35,54], which has been used in large scale cross-cultural studies [44,55,56]. This measure assesses six domains of flourishing: (1) happiness and life satisfaction, (2) mental and physical health, (3) meaning and purpose, (4) character and virtue, (5) close social relationships, and (6) financial and material stability. Each domain comprises two Likert scale questions, with each question’s response ranging from 0–10 (e.g., 0 = Extremely disagree and 10 = Extremely agree). VanderWeele and colleagues suggested that two summary flourishing scores can be generated. The “Flourish Index (FI)” is the average of scores from each of the first five domains which indicates flourishing at a given time [35]. The “Secure Flourish Index (SFI)” is the average of scores from all six domains which may indicate flourishing over an extended period [35]. Both indices’ average score ranges from 0–10, with higher scores indicating respondents perceive themselves more positively in terms of human flourishing. In a previous study with Chinese clothing supply chain workers, the Chinese version of FI and SFI had shown good internal consistency (Cronbach’s α = 0.88 and 0.81, respectively; [56]). The internal consistency of FI and SFI in this study sample was 0.91 and 0.89, respectively.

#### 2.3.4. Other Covariates

Demographic characteristics including gender (female vs. male), age (18–35 years), year in university (freshman, sophomore, junior, senior, and graduate school), and marital status (single, married or cohabitating, divorced, separated, widowed, and other) were also collected.

### 2.4. Data Analysis

Statistical analyses were performed using SPSS 27.0 [57]. Descriptive statistics (i.e., means, standard deviations [SDs], percentage, and frequencies) were used to describe study variables. Missing data patterns were assessed using the Expectation-maximization (EM) algorithm procedure and confirmed that data were missing at random. The EM procedure is a commonly used technique that finds maximum likelihood estimates for missing data [58]. To test the hypotheses that (a) greater exposure to PCEs will be associated with higher scores on flourishing indices and (b) higher levels of PCEs will attenuate the effect of ACEs on flourishing, multiple regression analysis was used to assess the impact of PCEs, ACEs, and interaction between PCEs and ACEs on flourishing indices. Step 1, ACEs total scores and all controlled variables were entered into the model. Step 2, PCEs total scores were entered into the model. Step 3, the interaction of ACEs and PCEs total scores was entered into the model to examine PCEs’ potential moderation effect. Covariates including gender, age, year in university, and marital status were controlled in all models. The goodness of model fit was examined using adjusted R^2^.

### 2.5. Ethical Approval and Informed Consent

The corresponding author’s university ethics institutional review board (IRB; Shanghai Jiao Tong University School of Medicine) approved this study (IRB Approval number: SJUPN-202004). Implied consent to participate was indicated when participants provided responses to survey items.

## 3. Results

### 3.1. Participants Characteristics and Flourishing

Our sample includes 9468 young adults with a mean age of 20.1 years (SD = 1.6). Most participants were undergraduate students (96.4%) and single (79.8%) and three-quarters of the participants were female (75.3%). The mean FI and SFI in our sample are 6.93 (SD = 1.65) and 6.87 (SD = 1.61), respectively. The means and standard deviations of all flourishing domains are shown in Table 1.

### 3.2. Prevalence of PCEs

The total PCEs score ranged from 0 to 9 (M = 8.38; SD = 1.25). A total of 1.16% (*n* = 110) reported 0–3 PCEs, 6.87% (*n* = 650) reported 4–6 PCEs, and 91.97% (*n* = 8708) reported 7–9 PCEs. Participants’ exposure to PCEs by items is presented in Table 2. “Family relationships are harmonious” (96.9%), “Feel supported by friends” (96.8%), and “Treated fairly at school” (96.3%) were the top three most frequently reported PCE items, followed by “Feel safe and protected by an adult in home” (95.6%) and “Receive affirmation, encouragement, or support” (94.7%). The least frequently reported PCE items were “Able to talk to your family about feelings” (86.4%) and “Feel a sense of belonging in school” (87.1%).

### 3.3. Correlations between PCEs, ACEs, and Adult Flourishing

Table 3 presents the correlations between the cumulative and subdimension scores of the PCEs, cumulative ACEs, and adult flourishing indices and domains. Both cumulative and subdimensions of PCEs had statistically significant positive correlations with all flourishing indices and domains. Cumulative PCEs had moderate correlations (r = 0.31–0.39) with all flourishing indices and domains except for a small correlation (r = 0.24) with domain 6, financial and material stability. PCEs household subdimension had small correlations (r = 0.19–0.28) with all flourishing indices and domains. PCEs community subdimension had moderate correlations (r = 0.30–0.36) with all flourishing indices and domains, except for small correlations with domain 4, character and virtue (r = 0.29), and domain 6, financial and material stability (r = 0.21). Cumulative ACEs had statistically significant negative correlations with all flourishing indices and domains (r = −0.22–−0.31) as well as with cumulative and both domains of PCEs (r = −0.28–−0.46).

### 3.4. PCEs Moderates ACEs’ Influences on Adult Flourishing

Table 4 presents the unstandardized regression coefficients of (a) cumulative ACEs scores, (b) cumulative PCEs scores, and the (c) interaction of cumulative ACEs and PCEs scores on flourishing indices across three models. The interaction term for cumulative ACEs and PCEs scores was statistically significant for both flourishing (b = −0.060 [−0.071, −0.049], *p* < 0.001) and secure flourishing (b = −0.057 [−0.068, −0.046], *p* < 0.001) indices. Figure 1 shows the influences of cumulative ACEs score on flourishing and secure flourishing indices, respectively, by levels of PCEs. Other co-variates were controlled in the Figure. As the number of ACEs increases, the decreases in flourishing and secure flourishing indices are smaller for those with higher levels of PCEs than those with lower levels PCEs. However, PCEs’ moderating impact gradually reduces as the number of ACEs increases and no longer has a statistically significant moderating impact after ACEs score reaches above nine.

## 4. Discussion

Our study examined the prevalence of PCEs and the associations between ACEs, PCEs, and levels of flourishing in a community sample of Chinese young adults. To our knowledge, this is the first study that (1) examined the relationship between PCEs and adult flourishing in a non-Western context and (2) explored PCEs’ moderating role in the relationship between ACEs and adult flourishing. Prior studies on PCEs have revealed that in the context of ACEs, higher levels of PCEs are associated with lower levels of psychopathology symptoms and health risks [11,12,13,25,27,29]. Our study extends the science by examining whether PCEs buffer the impact of ACEs on adult flourishing, a positive outcome and an indicator for overall well-being. Findings of our study have implications for research on child development and prevention interventions designed to promote resilience and mitigate the impact of childhood adversity on health outcomes and well-being.

The majority (91.97%) of our study participants reported high levels of PCEs (7–9 PCEs). This is higher than rates of PCEs reported in prior studies with Chinese samples, such as Chinese primary and secondary school students (*n* = 6363, 26.4% reported 6–7 PCEs [19]), Chinese elementary school students (*n* = 2288, 10.8% reported 4–5 PCEs [20], and Hong Kong university students (*n* = 332, 24% reported 9–10 PCEs [59]). Rates of PCEs in our sample are also higher than those reported in studies with US samples, including adults in Wisconsin (*n* = 6188, 52.3% reported 6–7 PCEs [13]) and juvenile offenders in Florida (*n* = 28,048, 31.97% reported 6 or more PCEs [17]). The variations in rates of reported PCEs can be explained by the differences in sample size, participants’ characteristics, and PCEs measures used across studies. For example, Qu and colleagues as well as Bethell and colleagues (2019) used the 7-item PCE measure [13,19]. Zhang and colleagues (2021) developed a 5-item PCEs index based on the presence or absence of five components, including high parental education, high perceived socioeconomic status, high parental warmth, two-parent family, and high peer support, to measure PCEs [20]. Whereas Xu and colleagues [59] used the Chinese version of the 10-item Benevolent Childhood Experiences scale [11] with items pertain to perceived relational and internal safety and security, positive and predicable quality of life, and interpersonal support. Thus, the differences in PCEs rates should be interpreted with caution given the inconsistencies of PCEs measures used across studies. Further, Wentzel et al. (2021) found that PCEs, such as support from peers, parental support, and school connectedness, are positively associated with academic achievement [60]. Our study sample consisted of Chinese undergraduate and graduate students, a group of young adults with high academic achievement. Thus, the high rates of PCEs in our sample may also reflect the characteristics of study participants. Finally, another reason our rates are higher may be how we dichotomized the measure responses and that we included “Sometimes” as an affirmative response (1 = Very often, often, and sometimes, 0 = Never or rarely) in the dichotomization, compared with the original PCEs study with US adults (Bethell et al., 2019), in which “Sometimes” was considered as a negative response (0 = Never, rarely, or sometimes).

Harmonious family relationship was the most frequently reported PCE in our sample. The salience of harmonious family relationships reflects the traditional Chinese culture and Confucian ideal, in which harmony in family and society is highly valued and aspired [47,48,61]. Lam and colleague’s (2012) qualitative study found that found that family harmony was the core element and prerequisite for family happiness and good family functioning in Chinese cultural context [61]. They posited that family harmony comprised four components: communication, mutual respect, lack of conflict, and family time [61]. Consistent with prior studies, our findings highlight the significance of harmonious family relationship in creating a supportive, sensitive, and nurturing child development environment in Chinese cultural context.

Being able to talk to family about feelings was the least reported household PCE in our sample. This may be explained by the persistent stigma on talking about emotions and mental health in Chinese culture [62]. Chinese parents are generally more implicit and less warm in emotional expression, compared to parents from other countries where a more authoritative parenting style predominates (e.g., the United States), and Chinese parents tend to express love for their child through practical actions, such as providing material support [63]. This finding also provides some insight on why emotional neglect was the most frequently reported ACE among Chinese young adults [44]. If children are unable to communicate with family members about their feelings, their emotional needs are less likely to be acknowledged and met. This underscores the need to develop or adapt existing family-based interventions that include a focus on communicating and responding to emotions [64] for use with Chinese families. Further, it may be important to incorporate strategies for fostering harmonious relationships that are in line with Chinese culture and/or Confucian beliefs when developing or adapting family-based interventions for Chinese families.

Prior research with Western samples found that PCEs were associated with greater flourishing in adults [11,13]. Our study with Chinese young adult showed a consistent pattern, that is, cumulative PCEs had statistically significant positive correlations with adult flourishing. Adding to the existing research on PCEs, we found that while both household and community subdimensions of PCEs were positively correlated with all flourishing indices and domains, community PCEs seem to have a stronger correlation with adult flourishing than household PCEs in our sample. The community PCEs sub-dimension included multiple items focused on school and supportive interpersonal relationships established through school (e.g., support from friends and teacher). The stronger correlation between community PCEs and adult flourishing may, in part, be explained by the critical role of school for Chinese children and adolescent and the significance of school interpersonal relationships in child development, mental health, school adjustment, academic engagement, school satisfaction, and psychological well-being which may in turn promote flourishing in adulthood [65,66,67,68,69].

In a highly competitive education system, Chinese students experience tremendous academic pressure and spend most of their time in school and school-based activities [70]. On average, Chinese secondary school students spend about 245 days per year in school. This means that Chinese children may spend equivalent, if not more, time with their peers and teacher at school than with their parents at home. Our findings regarding community-based PCEs highlight the opportunity to promote adult flourishing by implementing school-based interventions during the nine-year compulsory education in China that support positive peer and teacher-student relationships, particularly for those lacking a positive, nurturing environment at home. Prior research has identified a number of effective school-based interventions designed to promote student-student and student-teacher relationships that could be adapted for use in China’s compulsory education system (e.g., [71,72]).

In line with existing studies in the Western context that demonstrate positive experiences in childhood moderate adaptation in later life in the context of early life adversity [13,21,22,23], our key finding shows that PCEs buffer the deleterious impact of ACEs on adult flourishing among Chinese young adults. Our findings support the new conceptualization of Traumatic and Adverse Childhood Experiences (TRACES+; [31]) by providing empirical evidence on how protective factors, such as PCEs measured in this study, can help ascend individuals exposed to ACEs toward the pinnacle of resilience. More importantly, we found that PCEs’ protective role diminishes as ACEs increases and no longer has a statistically significant protective role when individuals reported exposure to nine or more ACEs. This affirms the importance of conducting routine screening for ACEs and PCEs simultaneously in the context of important childhood settings, such as primary medical care and compulsory education, so that individuals and groups with heightened ACEs burden can be identified and receive timely prevention and intervention efforts that mitigate the negative sequelae of ACE exposure and leverage their strengths [23,73]. In addition, similar to the National Survey of Children’s Health or state Behavioral Risk Factor Surveillance Surveys in the US, it will be important to develop comprehensive ACEs and PCEs public health surveillance strategies in China, in order to monitor the success of prevention and intervention efforts in reducing both ACE exposure and the negative consequences of such exposure.

### Limitations

First, this study used a cross-sectional design. Therefore, we cannot infer a causal relationship between ACEs, PCEs, and adult flourishing. A longitudinal design might be merited for future study to delineate the joint impact of adverse and positive childhood experience on adult health and well-being. Second, we elicit childhood experiences through participant self-report which is subject to recall bias and shared method bias. Future studies that include multiple informants and sources of data (e.g., self-report, parent/teacher-report, and clinical record) may be able to address this limitation. Third, our convenience sample constitutes relatively well-educated Chinese young adults who are college or graduate students. The high prevalence of PCEs may not be representative in the general Chinese adult populations, especially those with lower socioeconomic status. ACEs are common but disproportionately affect individuals in low-resourced communities who also tend to have fewer protective factors. As depicted in the new conceptualization of TRACE+ [31], both risks and protective factors exist in the context of and interact with historical, pre-existing, and contextual factors. From an equity perspective, future studies on ACEs and PCEs in China should expand to include groups with a broader range of socioeconomic statuses, especially those residing in low-resourced settings such as individuals living in rural areas ridden by poverty or in institutional orphanage, or those with disabilities. As such, resources can be effectively allocated to support developing or adapting and implementing inventions tailored to the needs of these groups and help to move them toward the pathway of resilience. Our PCE measure did not comprise a comprehensive list of all protective childhood experiences and may not adequately include other PCEs relevant to Chinese cultural context. Finally, we dichotomized the PCEs measure which may inflate the rate of PCEs. Although we believe that our modified lower threshold for a positive response on the PCEs measure is appropriate for our study sample given the documented cultural differences in both survey response patterns and parenting styles, we acknowledge that our modification may be too blunt to assess some hidden nuance that a more granular approach would otherwise reveal. Future study may consider evaluating how different scoring method may affect PCEs prevalence as well as its protective role in the context of ACEs exposure.

## 5. Conclusions

Our findings indicate that PCEs are common among Chinese young adults, with harmonious family relationships, feeling supported by friends, and being treated fairly at school being the most frequently reported PCEs in this sample. Individual PCE dimensions as well as the cumulative number of PCEs were positively correlated with multiple domains of flourishing. Moreover, PCEs served to buffer against the cumulative impact of ACE exposure on flourishing up to the point of experiencing eight ACEs. Given our findings that PCEs are not only positively correlated with flourishing but may also protect against the negative impacts of ACE exposure on flourishing, it is critical that we focus our efforts on both preventing or reducing ACE exposure and promoting PCEs beginning in early childhood and continuing throughout children’s development. Thus, there is a need for a comprehensive approach to promote the well-being of children and families that includes: (1) systematic screening of children and parents for both ACEs and PCEs; (2) connecting those who have significant exposure to ACEs with appropriate evidence-based treatments designed to mitigate the negative sequelae of ACE exposure and promote mental health and resilience (e.g., Child-Parent Psychotherapy; Trauma-Focused Cognitive Behavioral Therapy [74,75]); and (3) integrating efforts to promote PCEs into community settings such as early childhood and compulsory education and primary care settings where families spend a significant amount of their time. These efforts should include preventive interventions designed to strengthen parenting skills and parent-child communication and relationships, bolster family’s economic well-being through occupational training and employment services, and strengthen student-student and student-teacher relationships through schoolwide school climate interventions [71,76,77].

## Figures and Tables

**Figure 1 ijerph-19-14956-f001:**
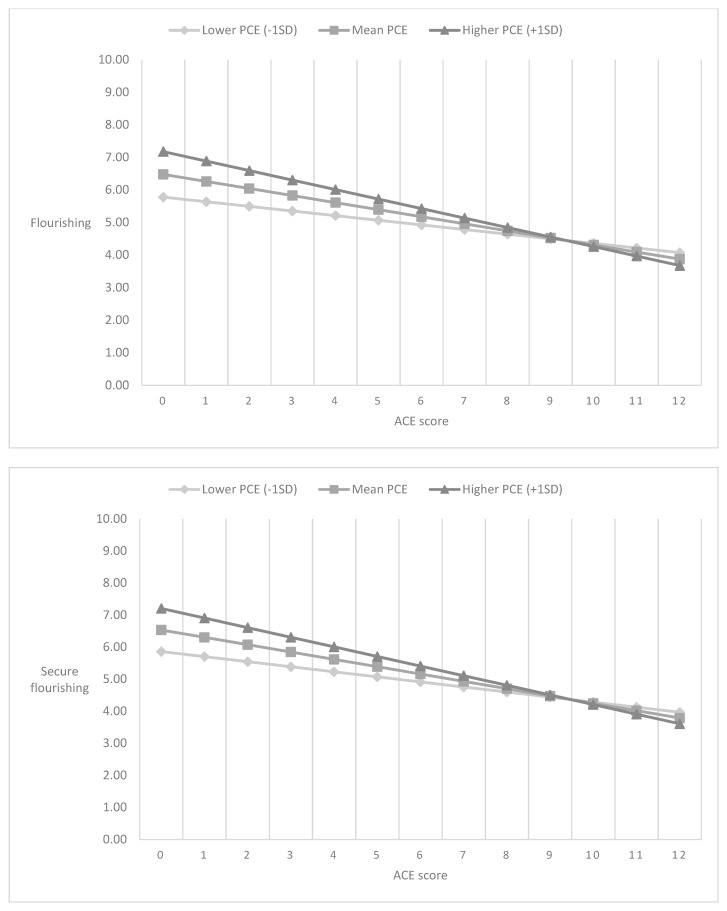
Flourishing indices by the cumulative ACEs scores and PCEs level.

**Table 1 ijerph-19-14956-t001:** Participant characteristics (*n* = 9468).

**Age (in years)**	
Range	18–35
Mean (SD)	20.1 (1.6)
**Gender, *n* (%)**	
Female	7129 (75.3)
Male	2244 (23.7)
Missing	95 (1.0)
**Year in university, *n* (%)**	
Freshman	2146 (22.7)
Sophomore	2652 (28.0)
Junior	2986 (31.5)
Senior	1342 (14.2)
Graduate	259 (2.7)
Missing	83 (0.9)
**Marital status**	
Single	7554 (79.8)
Married or cohabitate	107 (1.1)
Other *	1807 (19.1)
**Flourishing Measures, mean (SD)**	
Flourish Index (FI)	6.93 (1.65)
Secure Flourish Index (SFI)	6.87 (1.61)
Domain 1: Happiness and life satisfaction	6.91 (1.96)
Domain 2: Physical and mental health	7.50 (1.80)
Domain 3: Meaning and purpose	6.90 (1.92)
Domain 4: Character and virtue	6.67 (1.89)
Domain 5: Close social relationships	6.70 (1.99)
Domain 6: Financial and material stability	6.55 (2.42)

Note. * Other includes missing, divorced, separated, widowed, or other marital status. The “Flourish index” is the average of the first five domains. The “Secure flourish index” is the average of all six domains.

**Table 2 ijerph-19-14956-t002:** PCEs exposures by items.

**Household PCE, Mean = 0.93 SD = 0.18**	** *n* **	**%**
1. Able to talk to your family about feelings		
Never or rarely	1278	13.5
Very often, often, or sometimes	8178	86.4
Missing	12	0.1
2. Family stood by you during difficult times		
Never or rarely	548	5.8
Very often, often, or sometimes	8901	94.0
Missing	19	0.2
3. Feel safe and protected by an adult in your home		
Never or rarely	401	4.2
Very often, often, or sometimes	9048	95.6
Missing	19	0.2
4. Family relationships are harmonious		
Never or rarely	267	2.8
Very often, often, or sometimes	9172	96.9
Missing	29	0.3
**Community PCEs, mean = 0.93, SD = 0.15**	** *n* **	**%**
5. Treated fairly at school		
Never or rarely	293	3.1
Very often, often, or sometimes	9119	96.3
Missing	56	0.6
6. Feel a sense of belonging in school		
Never or rarely	1178	12.4
Very often, often, or sometimes	8250	87.1
Missing	40	0.4
7. Feel supported by friends		
Never or rarely	294	3.1
Very often, often, or sometimes	9163	96.8
Missing	11	0.1
8. Have at least 2 nonparent adults who took genuine interest in you		
Never or rarely	892	9.4
Very often, often, or sometimes	8524	90.0
Missing	52	0.5
9. Receive affirmation, encouragement, or support		
Never or rarely	481	5.1
Very often, often, or sometimes	8965	94.7
Missing	22	0.2

Note. PCEs = Positive Childhood Experiences.

**Table 3 ijerph-19-14956-t003:** Correlations between the cumulative and subdimension scores of the 9-item PCEs, cumulative ACEs, and adult flourishing indices and domains (all *p*-value < 0.001).

	Flourish Index	Secure Flourish Index	Domain 1: Happiness and Life Satisfaction	Domain 2: Physical and Mental Health	Domain 3: Meaning and Purpose	Domain 4: Character and Virtue	Domain 5: Close Social Relationships	Domain 6: Financial and Material Stability	ACEs Cumulative
PCEs cumulative	0.384 ** (0.366, 0.401)	0.387 ** (0.370, 0.404)	0.334 ** (0.316, 0.352)	0.339 ** (0.321, 0.357)	0.317 ** (0.298, 0.335)	0.306 ** (0.288, 0.324)	0.350 ** (0.333, 0.368)	0.237 ** (0.218, 0.256)	−0.439 ** (−0.455, −0.422)
PCEs Household	0.268 ** (0.250, 0.287)	0.276 ** (0.257, 0.294)	0.231 ** (0.211, 0.250)	0.250 ** (0.231, 0.269)	0.218 ** (0.199, 0.237)	0.219 ** (0.200, 0.238)	0.235 ** (0.216, 0.254)	0.186 ** (0.166, 0.205)	−0.457 ** (−0.473, −0.441)
PCEs Community	0.365 ** (0.347, 0.382)	0.363 ** (0.346, 0.381)	0.319 ** (0.300, 0.337)	0.312 ** (0.294, 0.330)	0.302 ** (0.284, 0.320)	0.289 ** (0.270, 0.307)	0.342 ** (0.324, 0.360)	0.208 ** (0.188, 0.227)	−0.284 ** (−0.302, −0.265)
ACEs cumulative	−0.293 ** (−0.312, −0.275	−0.306 ** (−0.324, −0.288)	−0.249 ** (−0.268, −0.230)	−0.309 ** (−0.327, −0.291)	−0.229 ** (−0.248, −0.209)	−0.226 ** (−0.245, −0.207)	−0.250 ** (−0.269, −0.231)	−0.223 ** (−0.242, −0.204)	-- --

Note. ACEs = Adverse Childhood Experience. PCEs = Positive Childhood Experiences. ** Correlation is statistically significant at the 0.01 level (2-tailed).

**Table 4 ijerph-19-14956-t004:** Relationships among Flourishing Indices, ACEs, and PCEs.

Explanatory Variables	Step 1	Step 2	Step 3
	*b* (2-Tailed 95% CI) *p*-Value	*b* (2-Tailed 95% CI) *p*-Value	*b* (2-Tailed 95% CI) *p*-Value
	Flourishing Index		
ACEs	−0.338 (−0.361, −0.315) <0.001	−0.181 (−0.205, −0.156) <0.001	0.286 (0.197,0.374) <0.001
PCEs		0.414 (0.386, 0.441) <0.001	0.561 (0.523, 0.599) <0.001
ACEs X PCEs (interaction term)			−0.060 (−0.071, −0.049) <0.001
Adjusted R^2^	0.092	0.172	0.183
	Secure Flourishing Index		
ACEs	−0.344 (−0.367, −0.322) <0.001	−0.192 (−0.216, −0.168) <0.001	0.249 (0.163, 0.335) <0.001
PCEs		0.401 (0.374, 0.428) <0.001	0.540 (0.503, 0.578) <0.001
ACEs X PCEs (interaction term)			−0.057 (−0.068, −0.046) <0.001
Adjusted R^2^	0.100	0.179	0.189

Note. All models controlled for age, gender, and year in university; ACEs = adverse childhood experience.

## Data Availability

Due to the sensitive nature of the questions asked in this study, survey respondents were assured raw data would remain confidential and would not be shared.

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
