# Peer review of "Positive Childhood Experiences Associate with Adult Flourishing Amidst Adversity: A Cross Sectional Survey Study with a National Sample of Young Adults"

_ijerph, 2022, doi:10.3390/ijerph192214956_

Round 1

Reviewer 1 Report

Thank you for the opportunity to review this manuscript. Strengths include a focus on an important but understudied topic (PCEs), setting in mainland China (given that most literature has been conducted in Western countries), and a relatively large sample size (~9400). The manuscript content is comprehensive, clear, and well-written. It is a descriptive study, which limits its impact, but is appropriate given the early state of the research in this specific area. Opportunities for refinement, for the authors’ consideration, are detailed below. My suggestions are minimal, as the manuscript is of high quality in its current state.

Introduction

*line 85 – Can the authors briefly introduce resilience theory? The readership of IJERPH is diverse and broad, and thus a brief 1-2 sentence overview would be of value to those outside the field.

*line 85-94 – Can the authors briefly mention hypothesized mechanisms for PCE’s beneficial effects? I understand the literature is limited, but even a brief summary or 1-2 examples of how they are hypothesized to interrupt ACEs’ physiological/behavioral/psychosocial effects would be of value.

Methods

*Can the authors clarify the “EM procedure” for missing data? (line 227)

Results

*The authors modified the PCE questionnaire to combine “sometimes” with “very often” and “often,” to indicate a PCE exposure. This led to a prevalence of >~85% for all PCEs (and >~95% for several PCEs). It is possible that their modification was too blunt to assess nuance (especially given existing critiques of assessing ACEs/PCEs exposures in a binary manner). If their measurement approach was accurate, PCE exposure is so common to be almost a universal exposure in this sample; if it was not accurate, a more granular measurement would be needed to better assess hidden nuance. Can the authors address this in greater depth in their discussion or limitations? (Currently, it is very briefly mentioned in the discussion.)

*Can the authors clarify whether Figure 1 presents the results of controlled analyses?

Discussion

*The authors mention in the limitations that their sample was one of relative privilege/success (eg, university students). However I would like to see more depth in the discussion about the implications of this, especially because socially marginalized or more “vulnerable” populations often bear a disproportion burden of higher ACEs and lower PCEs in other countries. It seems there is a need, from an equity perspective, to explore their research questions in more vulnerable samples. I would be interested in the authors’ thoughts on this, include how it would be relevant to equity issues in China.

Reviewer 2 Report

This is an interesting idea and fits provides evidence of a new conceptualization of ACEs  Weems, C.F., Russell, J.D., Herringa, R.D., & Carrión, V. G., (2021). Translating the neuroscience of adverse childhood experiences to inform policy and foster population level resilience. American Psychologist, 76(2), 188–202. DOI: 10.1037/amp0000780. Presented an new ACEs pyramid which emphasizes how protective factors can help move individuals exposed to ACEs towards resilience - the Model in Weems et al would add support for the rationale for the study and help contextualize findings in the discussion.

In terms of the moderation effect the authors plot the effect but don't provide follow up analyses to decompose the interaction simple slopes analysis is needed in this case I would like to see the results of the The Johnson-Neyman procedure. This can be used to identify the point(s) along a continuous moderator where the relationship between the independent variable and the outcome variable transition(s) between being statistically significant to nonsignificant and is particularly interesting in this case and will aid the interpretation presented in the results and discussion section.
